# Evaluation of a comprehensive health check offered to frontline health workers in Zimbabwe

Edson T. Marambire[1,2,3☯]*, Rudo M. S. Chingono[1☯], Claire J. Calderwood[1,4],
Leyla Larsson[1,3], Sibusisiwe Sibanda[1], Fungai Kavenga[1,5], Farirai P. Nzvere[1,4], Ioana
D. Olaru[1,4], Victoria Simms[1,6], Grace McHugh[1], Tsitsi Bandason[1], Nicol Redzo[1], Celia
L. Gregson[1,7], Aspect J. V. Maunganidze[8], Christopher Pasi[9], Michael Chiwanga[10],
Prosper Chonzi[11], Chiratidzo E. Ndhlovu[12], Hilda Mujuru[13], Simbarashe Rusakaniko[14],
Rashida A. Ferrand[1,4], Katharina Kranzer[1,3,4]

1 The Health Research Unit Zimbabwe, Biomedical Research and Training Institute, Harare, Zimbabwe,
2 CIH[LMU] Center for International Health, University Hospital, LMU Munich, Munich, Germany, 3 Division of
Infectious Diseases and Tropical Medicine, Medical Center of the University of Munich, Munich, Germany,
4 Clinical Research Department, London School of Hygiene & Tropical Medicine, London, United Kingdom,
5 National TB Programme, Ministry of Health and Child Care, Harare, Zimbabwe, 6 International Statistics
and Epidemiology Group, London School of Hygiene & Tropical Medicine, London, United Kingdom, 7 Global
Health and Ageing, Bristol Medical School, University of Bristol, Bristol, United Kingdom, 8 Parirenyatwa
Hospital, Harare, Zimbabwe, 9 Sally Mugabe Central Hospital, Harare, Zimbabwe, 10 Chitungwiza Hospital,
Chitungwiza, Zimbabwe, 11 Harare City Health, Harare, Zimbabwe, 12 Internal Medicine Unit, University of
Zimbabwe College of Health Sciences, Harare, Zimbabwe, 13 Department of Paediatrics and Child Health,
University of Zimbabwe College of Health Sciences, Harare, Zimbabwe, 14 Department of Community
Medicine, College of Health Sciences, University of Zimbabwe, Harare, Zimbabwe

☯ These authors contributed equally to this work.
* Edson.Marambire@lrz.uni-muenchen.de

pgph.0002328

Western Cape, SOUTH AFRICA

**Data Availability Statement:** Data has been
uploaded and openly available on the London
School of Hygiene and Tropical medicine (LSHTM)

## Abstract

Health workers are essential for a functioning healthcare system, and their own health is
often not addressed. During the COVID-19 pandemic health workers were at high risk of
SARS-CoV-2 infection whilst coping with increased healthcare demand. Here we report the
development, implementation, and uptake of an integrated health check combining SARS-
CoV-2 testing with screening for other communicable and non-communicable diseases for
health workers in Zimbabwe during the COVID-19 pandemic. Health checks were offered to
health workers in public and private health facilities from July 2020 to June 2022. Data on
the number of health workers accessing the service and yield of screening was collected.
Workshops and in-depth interviews were conducted to explore the perceptions and experi-
ences of clients and service providers. 6598 health workers across 48 health facilities
accessed the service. Among those reached, 5215 (79%) were women, the median age
was 37 (IQR: 29–44) years and the largest proportion were nurses (n = 2092, 32%). 149
(2.3%) healthcare workers tested positive for SARS-CoV-2. Uptake of screening services
was almost 100% for all screened conditions except HIV. The most common conditions
detected through screening were elevated blood pressure (n = 1249; 19%), elevated HbA1c
(n = 428; 7.7%) and common mental disorder (n = 645; 9.8%). Process evaluation showed
high acceptability of the service. Key enablers for health workers accessing the service
included free and comprehensive service provision, and availability of reliable point-of-care

data compas. https://doi.org/10.17037/DATA. 00003679.

**Funding:** This work was supported by the Global Public Health strand of the Elizabeth Blackwell Institute for Health Research, funded under the University of Bristol's QR GCRF strategy (ref: H100004-148) as well as support from Sheffield and Oxford QR-GCRF funds. It was supported by UK aid from the UK government (FCDO) (ref 668 303), and by funding from the government of Canada; the views expressed do not necessarily reflect the policies of the respective governments. RAF is funded by a Wellcome Trust Senior Fellowship (206316_Z_17_Z). IDO and CJC have received funding through the Wellcome Trust Clinical PhD Programmememe awarded to the London School of Hygiene & Tropical Medicine (grant number 203905/Z/16/Z). VS received funding from the UK Medical Research Council (MRC) and the UK Foreign, Commonwealth and Development Office (FCDO) under the MRC/FCDO Concordat agreement which is also part of the EDCTP2 programmememe supported by the European Union (grant number MR/R010161/1). The funders had no role in study design, data collection and analysis, decision to publish, or preparation of the manuscript.

**Competing interests:** The authors have declared that no competing interests exist.

screening methods. Implementation of a comprehensive health check for health workers was feasible, acceptable, and effective, even during a pandemic. Conventional occupational health programmes focus on infectious diseases. In a society where even health workers cannot afford health care, free comprehensive occupational health services may address unmet needs in prevention, diagnosis, and treatment for chronic non-communicable conditions.

## Introduction

Delivery of high quality health care relies on adequate numbers of well trained, accessible and motivated health workers covering clinical and non-clinical roles [1]. Health workers are essential to functioning healthcare systems, improving the health of the population, achieving universal health coverage (UHC), and the sustainable development goals (SDGs). Globally countries are facing, to varying degrees, difficulties in the education, employment, deployment, retention, and performance of their health workforce [2, 3].

The World Health Organization (WHO) predicts a deficit of 18 million health workers worldwide by 2030 [4]; with a deficit of 6.1 million in the WHO Africa region alone. Africa has the lowest number of health workers overall, the lowest density of health workers per 1000 population, yet the highest disease burden [3, 5]. The average density of physicians, nurses, and midwives is 1.55 per 1000 population, far below the WHO SDG threshold of 4.45 health workers per 1000 population needed to achieve UHC [6]. Factors contributing to the health workforce crisis in the region include inadequate training capacity, demographic conditions (rapid population growth leading to more demand for more health workers), international migration, an ageing workforce, career changes, poor retention, morbidity, and premature mortality.

The COVID-19 pandemic has thrown a stark light on the vital role of health workers in ensuring health systems continue to function during times of crisis. The pandemic increased demands for health services, exceeding the capacity of health workers and resulting in burnout. It put health workers' health, well-being, and security at risk as they faced worsening working conditions, violence, harassment and increased risk of SARS-CoV-2 infection [7, 8].

Zimbabwe has about 50 000 health workers, with more than half being nurses [3], and most working in the public sector. As a result of the fragile economic climate and underfunding of the health system, health workers in Zimbabwe are poorly remunerated and work under difficult conditions [9, 10]. The limited data available suggest that undiagnosed chronic diseases such as diabetes, anaemia, and hypertension are highly prevalent in Zimbabwe [11, 12]. These conditions likely affect healthcare workers to a similar extent as the rest of the population. Diagnosing and treating chronic conditions has the potential to enhance healthcare workers' well-being. In the context of COVID-19, this is even more important because undiagnosed and poorly controlled chronic conditions are risk factors for severe COVID-19 [13]. Health workers, as well as the general population, mostly fund healthcare expenditures out of their own pocket, resulting in challenges in accessing health services, particularly for chronic conditions [14, 15].

At the start of the SARS-CoV-2 pandemic, health workers in Zimbabwe felt particularly vulnerable [16], resulting in industrial action over difficult working conditions, a lack of personal protective equipment (PPE), and poor remuneration. In July 2020 we set up a comprehensive health check offering testing for SARS-CoV-2 and screening for other communicable and non-communicable diseases (NCDs) for frontline health workers.

Here we report how the health check service was developed, implemented, and adapted, incorporating client feedback. We describe the geographical reach, uptake and yield of individual components of the service, and discuss recommendations for future implementation.

## Methods

### Study setting and population

A comprehensive health check programme was implemented in health facilities including tertiary-level hospitals, provincial and district (secondary-level) hospitals, primary health clinics, private and mission hospitals, in Zimbabwe between 29 July 2020 and 31 July 2022. Health facilities were chosen on the basis of need (high rates of SARS-COV-2 cases or expression of need for the services), and what was logistically feasible to implement by an organisation based in Harare, taking into consideration the impact of SARS-COV-2 travel restrictions on transport and working hours, especially during national lockdowns.

Prior to offering services at a health facility, permission was sought from the responsible authorities including provincial health directors, city health authorities, and institutional managers. Information about the service was shared with department heads, the principal nursing officer and/or matrons, as well as infection, prevention and control teams with the aim of raising awareness of the service among staff. Poster and flyers at hospital sites were used to publicise the service (S1 Fig). Service provision started in July 2020, during a period of SARS-CoV-2 related lockdown in Zimbabwe, which meant service operating hours were restricted to 9:00–14:00 (Fig 1). Initially, a team of two to four nurses and two to four assistants offered services at a single health facility. Between October 2020 and January 2021, the team was split into two, each with two nurses and two assistants providing services at two facilities in parallel: one of the teams was mobile and rotated through primary health clinics. Following the second

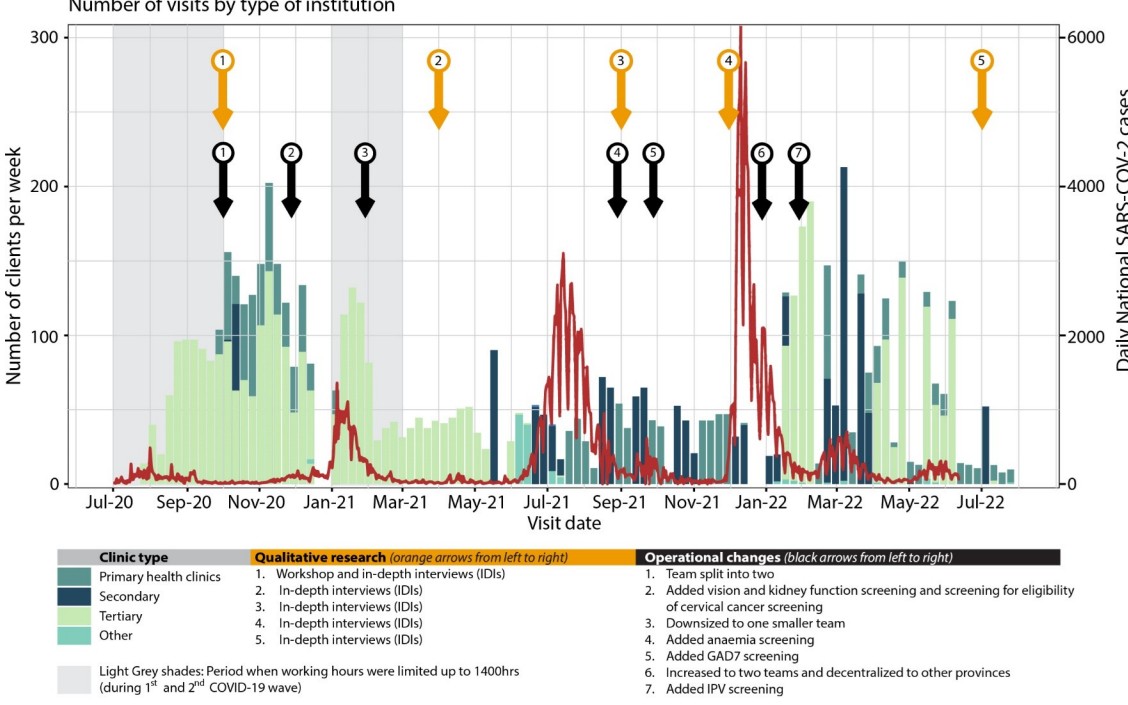

**Fig 1. Weekly enrolment, notified national SARS-CoV-2 infections and service development.**

SARS-CoV-2 wave in February 2021, the team was downsized with services provided by one team in one health facility at a time. In January 2022, a second team was employed to allow for service provision outside of Harare as part of the decentralisation of COVID-19 care in the country.

Depending on the size of the facility, services were offered for several days, weeks, or months until saturation was reached (i.e., until the service was being accessed by fewer than five clients per day). In health facilities beyond daily commuting distance from Harare, services were offered for 10 successive days (including weekends) with the study team residing on-site. In some health facilities in and around Harare, the service was offered over two time periods that were a minimum of six months apart, to allow for repeat check-ups.

All staff who were working at a health facility where the service was offered were eligible, regardless of their role and employer.

## Service setup

The service adhered to WHO SARS-CoV-2 infection prevention and control measures [17], including those for SARS-CoV-2 sampling. Workstations were set up outdoors using tents, or in dedicated rooms with open windows to ensure good ventilation. The team always encouraged social distancing and wore masks. Surgical masks were provided to clients.

The flow of a client through the service is described in Fig 2. Clients were free to choose any or all the services on offer. The service started at the waiting area [1] where potential clients were briefed about the services and given health information and an information sheet describing available services. Those who were interested proceeded to the registration tent [2] where they were asked for verbal consent, screened for symptoms of common mental health disorders and blood pressure, weight, and height were measured by a study assistant. From

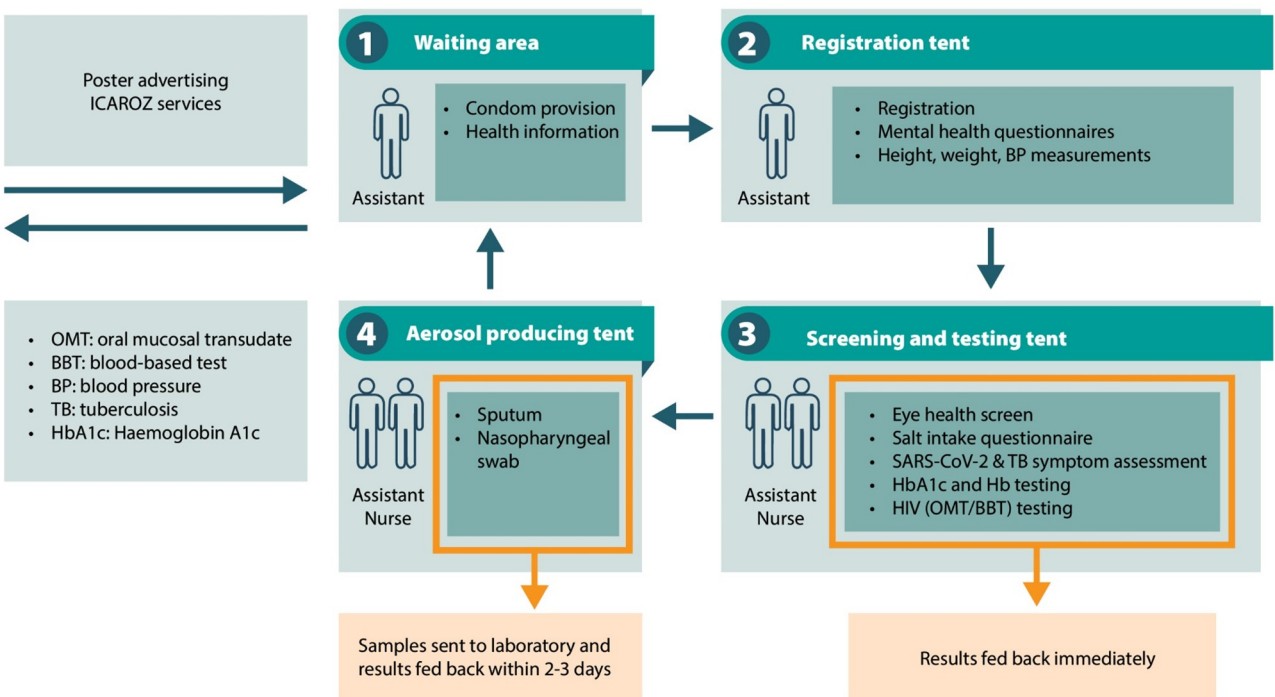

**Fig 2. Set-up of workstation and client flow through the occupational health program.**

there, they proceeded to the tent, staffed by a nurse [3], where all blood-based tests were offered with results fed back immediately (except for creatinine). The nurse discussed all screening results and offered referral where applicable. Those with respiratory symptoms proceeded to a tent designated for the aerosol producing procedures [4] for sputum sampling (tested for tuberculosis (TB) using Xpert MTB/Rif off-site) and nasopharyngeal swabs (investigated for SARS-CoV-2 by real-time Polymerase Chain Reaction (PCR) off-site). Clients would then complete the cycle at the waiting area where they would be offered free condoms and in addition, sanitary pads if they were female.

## Health checks

The health checks were offered on weekdays (except for facilities outside of Harare), during facility operating hours (9:00–15:00), and by appointment so as not to disrupt normal work activities at the health facilities. Walk-ins were accepted when numbers were low. The health checks were offered free of charge.

The initial composition of the health check was informed by ethical principles for screening, particularly that i) the conditions are of public health importance in Zimbabwe, ii) health workers are at increased risk, iii) a suitable point-of-care diagnostic test was available, and iv) there was an appropriate place to which to refer people who screened positive for further tests and/or treatment. The following conditions were included in the initial health checks: SARS-CoV-2, TB, HIV, hypertension, diabetes, and common mental health disorder. Table 1 describes eligibility for screening and associated screening tools, definition of a positive test screen, and steps for further assessment, management and care following a positive result.

Point-of-care diagnostic tests were used for all conditions except for SARS-CoV-2 and TB. Respiratory samples were collected and stored temporarily in insulated thermal bags before they were sent to laboratories (within 24 hours) and results fed back to clients within 12 to 36 hours. Health workers testing SARS-CoV-2 positive were called on day 1, 3, 7 and 10 after accessing the service to inform them about the result and check on their symptoms. Free SARS-CoV-2 testing was available for any family members of the health workers testing SARS-CoV-2 positive who had respiratory symptoms. The health check was iteratively developed based on ongoing feedback from both clients and providers (screening team), which resulted in expansion to include additional screening tests (kidney function, vision, anaemia, cervical cancer eligibility screening, and intimate partner violence) as part of the service at various time points (Fig 1).

## Referrals

Positive screening results (as defined in Table 1) were discussed with the clients and referral offered. Those who agreed to be referred were given a referral letter addressed to a provider of their choice. Clients found to have symptoms of common mental health disorders were asked for consent to be contacted by the Counselling Service Unit (CSU); a private voluntary organization which offers counselling services to clients in need. If they agreed, they were contacted by CSU by phone or WhatsApp and received counselling without any costs [28].

In January 2021, follow-up phone calls were conducted to establish and facilitate linkage to care for those who were referred for any condition.

## Quantitative data and analysis

Quantitative data were recorded on Samsung tablets using SurveyCTO software (Dobility, Inc), a secure electronic mobile data collection and management system loaded onto the tablets. These data consisted of screening results as well as the client's socio-demographic

**Table 1. Screened condition, screening tool, cut-points and referral pathways.**

| Condition | Eligibility* and screening tool | Definition of positive screen | Further assessment, treatment and care following positive screen |
|---|---|---|---|
| **Non-communicable diseases** | | | |
| **Mental Health** | Screening tool: audio computer-assisted self-interviewing using electronic tablets or paper-based questionnaires. Tools: Shona Symptom Questionnaire (SSQ-14), Generalized Anxiety Disorders assessment (GAD-7), WHO questionnaire for intimate partner violence (IPV) | SSQ-14 $\geq$ 8 or a 'red flag' (suicidal thoughts or hallucinations) and/or GAD-7 $\geq$ 10, Sexual abuse or moderate to severe physical abuse [18, 19] | Referred to Counselling Services Unit (CSU), Active contact by CSU and counselling by phone or WhatsApp. Referred to Musasa project if positive IPV screens: clients were offered a phone number and given the option to contact Musasa project. Clients were also counselled by the service team using the WHO LIVES approach. |
| **Eye Health** | Screening tool: Distance vision was assessed using the PEEK acuity app and near vision was assessed using a near vision chart | Score 6/12 or higher for distance and/or near vision on either eye [20] | Referred to local eye clinic (Council for the Blind Zimbabwe) for refraction and subsidized glasses if required |
| **Hypertension** | Screening tool: Blood pressure (BP) measured three times at intervals of at least five minutes with the lowest measurement recorded. | Systolic BP>140mmHg and/or diastolic BP>90mmHg [21] | Referral to health provider of the client's choice Malignant hypertension (systolic BP above 180mmHg) and hypertensive pregnant women referred to the emergency department |
| **Diabetes** | Screening tool: HbA1c measured using point of care Haemoglobin A1c machine (A1c Care, SD Biosensor, Singapore) | HbA1c $\geq$ 6.5% [22] | Referred to health provider of the client's choice |
| **BMI** | Screening tool: BMI, calculated from height and weight and categories according to WHO. | BMI <18.5kg/m$^2$ defined as underweight. BMI 25kg/m$^2$–30kg/m$^2$ defined as overweight BMI $\geq$30kg/m2 defined as obese [23] | Advised regarding nutrition and physical activity |
| **Anaemia** | Screening tool: Haemoglobin level (Hb) using HemoCue analyser (Hemocue 301, HemoCue, Sweden). | Hb < 8.0 g/dL (severe anaemia) [24] | Referred to health provider of the client's choice |
| **Cervical Cancer** | Eligibility for cervical cancer screening was assessed according to national guidelines. • PLHIV eligible if screened > 1 year ago • HIV- client eligible if >30 years and screened > 3 years ago [25] | | Anyone eligible for screening was referred to preferred clinic or hospital with a Visual Inspection with Acetic Acid and Cervicography (VIAC) department |
| **Kidney Disease** | Eligibility: Those with known but poorly controlled hypertension/diabetes or newly diagnosed diabetes Screening tool: laboratory-based creatinine testing | Electrolytes, urea, creatinine and eGFR results were received from the laboratory and interpreted using reference ranges provided by the laboratory | Results available within 48h, clients actively followed up for all results, those with abnormal creatinine or GFR were referred to health provider of the client's choice |
| **Infectious diseases** | | | |
| **HIV** | Eligibility: not known HIV positive & last HIV test > 3 months ago. Screening options: • Oral mucosal transudate (OMT) self-test kit (OraSure Technologies, USA), on or offsite test • Provider rapid blood-based test (BBT) (Alere Determine HIV 1/2, USA) + confirmatory BBT Chembio | Test positive for HIV 1 / 2 | Referred to health provider of the client's choice. Clients opting to test offsite were given information about local services in the case of a positive result but were not actively followed up for results. |
| **Respiratory infections (SARS-CoV-2, tuberculosis)** | Eligibility for SARS-CoV-2: Any respiratory symptoms: runny nose, sneezing, coughing, fever, headache, myalgia/arthralgia, fatigue, pharyngitis, diarrhoea, night sweats, skin rash, lymphadenopathy, oral ulcers or loss of smell or taste [26, 27] Eligibility for TB: WHO symptom screen positive Screening test: Nasopharyngeal swab (SARS-CoV-2) and/or sputum sample (TB) for laboratory testing | SARS-CoV-2: positive RT PCR TB: MTB DNA detected (Xpert MTB/Rif) | **For positive SARS-CoV-2 results:** clients were informed of results by telephone and provided with information and counselling regarding complications, at-risk household contacts and prevention of onward transmission. Follow-up calls were made 3, 7, and 10 days after the positive result to assess severity and facilitate medical care. Infection control and prevention team at health facilitates were informed about the results if the client consented. **For positive TB results:** clients were called and referred to a local TB clinic of their choice. **Negative results** were communicated by text message. |

information, medical history, SARS-CoV-2 risk perception and SARS-CoV-2 knowledge, attitudes, and practices. Medical history, including the following conditions, was ascertained: anaemia, asthma, cardiovascular disease, diabetes, epilepsy, HIV, hypertension, past or present malignancies, renal disease, past tuberculosis, confirmed SARS-CoV-2 infection, anxiety, and depression. Anthropometric measurements, uptake of screening and results of tests conducted on site were also recorded. Laboratory tests results were sent to the study lead, shared with the data team and captured through SurveyCTO. Recorded data were extracted from the SurveyCTO server and uploaded to a Microsoft SQL. The server was encrypted and hosted at the Biomedical Research and Training Institute.

Data analysis was performed using Stata (version 13) and R (version 4.2). Continuous data were summarized by their mean and standard deviation or median and interquartile ranges (IQR); categorical data were summarized as frequencies with percentages. Thresholds to define prevalence of diseases such as raised blood pressure, HbA1c, and underweight/overweight followed WHO guidelines [29, 30]. A proportional symbols plot was used to describe geographical coverage of the service.

## Qualitative data collection and analysis

We used qualitative methods to explore the perceptions and experiences of clients and service providers. We began by conducting two participatory research workshops (September–November 2020) aimed at providing insights into health worker health needs, and how best to structure the occupational health-check services. During the workshops, feedback was actively sought on what was working well and what could be improved. Findings from these workshops informed refinement of the intervention.

In-depth interviews (IDIs) were conducted by research assistants across five timepoints throughout the intervention (Fig 1). Health workers were purposively selected (gender, age, profession, and conditions for which they screened positive) and asked for their perceptions, experiences with, and suggestions for refinement of the intervention.

Two team meetings with staff providing the services were held in September and November 2020 to discuss progress and suggestions for improvement. Feedback from team members and notes from subsequent monthly staff meetings were reviewed as part of the process evaluation.

The qualitative data collection was undertaken by trained research assistants in English or in local languages (Shona or Ndebele) according to participant preference. IDIs were audio recorded, transcribed verbatim and, where needed, translated into English. Recordings from interviews were stored on an encrypted hard drive. Transcriptions, in which participant identifiable information had been removed, were uploaded to a secure cloud server accessible to authorized study staff only. Following checking of transcriptions for accuracy, audio recordings were deleted.

All qualitative data were uploaded onto NVivo 14 for coding and analysis. Analysis was conducted by a trained qualitative researcher (SS) and a social scientist (RC) guided by the principles of thematic analysis [31]. Six transcripts were used to create a coding framework which was refined as new themes and patterns emerged. Iterative thematic analysis was used on an ongoing basis to explore both deductive themes (identified before data collection) and inductive themes (emerging from the data). The main deductive themes were "Challenges of an under resourced health system", "Poor health seeking behaviours in health workers", "financial constraints" which fed into the results section on the barriers to engagement with the health check-ups. "Accessibility", "free", and comprehensive service provision" fed into the results section on motivation for engagement with the health check-ups. The main inductive themes were "good quality screening tools", "friendly staff", "privacy and confidentiality",

prioritizing health worker's health needs" which were identified as perceived benefits and positive attributes facilitating the engagement with the health check-ups. "Denial". "Time constraints" were identified as recuring themes identified and this fed into the barriers of service uptake. In addition to the inductive themes, "expansion of services", "repeat service provision" and "free referral services" were identified and these fed into adaptations and key recommendations for future interventions. Analysis was an ongoing process and as a result, some of the key findings and recommendations were used to continuously refine the ongoing intervention and inform data collection.

Analytical memos were drafted to explore the connection between codes and further develop emerging ideas, and to highlight and arrange significant themes. Coded excepts from the transcripts were grouped together under the identified themes presented in this paper (S1 Table).

## Ethical approval

Ethical approval was granted by the Medical Research Council of Zimbabwe (MRCZ/A/2627), and the London School of Hygiene & Tropical Medicine ethics committee (22514) prior to participant recruitment. The study was granted a waiver allowing for verbal rather than written consent by health workers accessing the health check-up since the primary aim of the project was to provide a service. Informed written consent was obtained for all qualitative workshops and interviews. The UK and Zimbabwean data protection regulations were adhered to.

## Results

### Service reach and results from the health checks

Between July 2020 and July 2022, comprehensive health checks were offered to health workers at 48 health facilities across all ten provinces of Zimbabwe (Fig 3). Among the facilities were four private health facilities and three faith-based (mission) health facilities. Health checks were offered over 407 days, with a median of 14 (IQR: 9–22) health workers accessing the service per day. A total of 6598 health workers accessed the service at least once, 1350 from primary, 1361 from secondary and 4073 from tertiary and quaternary-level facilities.

The number of team members working during SARS-CoV-2 waves was impacted by SARS-CoV-2 infections among the team and the resulting need for isolation. This resulted in fewer clients being served during periods of increased SARS-CoV-2 infection rates (Fig 1).

Overall, 5215 (79%) clients attending the service were women; and the overall median age was 37 (IQR: 29–44) years (Table 2). The largest occupation group accessing the service were nurses (n = 2092, 32%) and non-clinical support staff (n = 1880, 29%). Doctors, police or army personnel, and porters together contributed fewer than 3% of those who accessed the service.

A total of 3170 (48%) health workers reported they had previously been diagnosed with at least one condition. Hypertension (1124, 17%) was the most commonly reported condition, followed by SARS-CoV-2 (1109, 17%) and HIV (675, 10%) (Table 3). Uptake of each component of the services was almost 100% for all conditions except for HIV testing (Fig 4). Among health workers who did not have either a previous HIV diagnosis or reported a negative HIV test in the past 3 months, 3967/5568 (71.2%) took up HIV testing. The majority (54.5%) opted for blood-based testing by a service provider, whilst 925 (16.6%) took an oral mucosal self-test kit to test off-site. Screening revealed a high prevalence of elevated blood pressure (1249/6598, 19.0%) and increased HbA1c (428/5532, 7.7%) and symptoms of common mental health disorder (645/6598, 9.8%). The prevalence of symptoms suggestive of SARS-CoV-2 was 14% (920/6598) with a total of 149 people (2.3%) testing positive for SARS-CoV-2. Only 40 health workers screened positive on the WHO TB symptom screen [32], of which 31 submitted a sputum

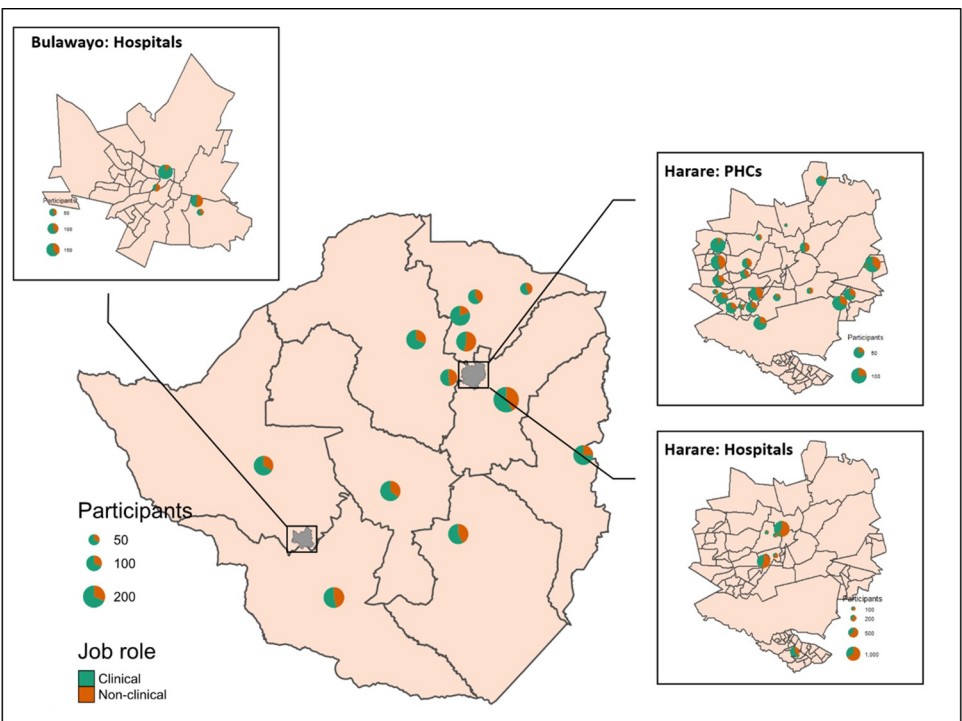

**Fig 3. Map of Zimbabwe showing the geographical spread of the service provision.** The map was created using shape files from Humanitarian Data Exchange; Zimbabwe National Statistics Agency, and plotted in R using the sf and tmap packages. https://data.humdata.org/dataset/cod-ab-zwe.

sample, and none of the samples tested positive for *Mycobacterium tuberculosis*. Vision impairment was found in 9.8% (452/4629). 878/5215 (16.8%) women were eligible for cervical cancer screening. A total of 17 (0.6% of those tested) had a new HIV diagnosis made through the health check.

## Motivation for engagement with the health check-up service

The process evaluation identified several attributes of the service which facilitated uptake. Health workers liked that the service was free, comprehensive, and well-equipped with reliable point-of-care screening methods. They appreciated that not only was it accessible, but it was a 'one-stop-shop' screening for multiple health conditions.

> *"Then the other thing is that you are coming to us. . .at times we don't create time to go for just a health check-up. We just then go for testing when we are already sick so I think it's helping us a lot. . ." (Nurse, Female, 30)*

> *"I would say it was quite an interesting experience. You know that moment when you get everything in one package. Getting your blood pressure checked, your eyesight, screened for COVID and even checked up on your mental health. . . having a one stop shop. It was a beautiful experience if I might say." (Nurse, female, 31)*

> *"I also liked the HbA1c test because it's different from getting tested (with a glucometer). . . it's a useful screening tool that can actually tell if someone is diabetic or not." (Sister-in-charge, female, 39)*

**Table 2. Characteristics of health workers screened through the comprehensive health check program; July 2020—July 2022, in the ICAROZ study.**

| Characteristics | | n = 6,598, n (%) |
|---|---|---|
| **Sex** | | |
| Male | | 1,383 (21.0%) |
| Female | | 5,215 (79.0%) |
| **Age** | | |
| 18–29 years | | 1,735 (26.3%) |
| 30–39 years | | 2,245 (34.0%) |
| 40–49 years | | 1643 (24.9%) |
| 50+ years | | 975 (14.8%) |
| **Employer** | | |
| Government health service | | 5713 (86.6%) |
| Private or mission hospital | | 338 (5.1%) |
| NGO/Other | | 547 (8.3%) |
| **Highest level of education achieved** | | |
| No school/primary | | 183 (2.8%) |
| O-levels | | 2,557 (38.8%) |
| A-levels | | 656 (10.0%) |
| Diploma after secondary school | | 2,384 (36.1%) |
| University | | 818 (12.4%) |
| **Occupation** | | |
| Clinical | Nurses | 2092 (31.7%) |
| | Clinical support staff | 1876 (28.4%) |
| | Allied health professionals | 407 (6.2%) |
| | Doctors / medical students | 186 (2.8%) |
| Non-clinical | Non-clinical support staff | 1880 (28.5%) |
| | Other* | 160 (2.4%) |

Abbreviations

NGO = non-governmental organisation

Footnotes

* Other category includes students and tutors

In addition to being comprehensive, health workers expressed how the efficiency and the quality of service (including privacy and friendly staff) drew people to seek services.

> *"The service was good, it was professional. The way they did their operations, everything was good, even when I was tested it didn't take time, I arrived and got service immediately." (Primary counsellor, Female, 41)*

> *"The service is good. We feel at ease coming here than the staff clinic because we know our privacy will be respected." (Nurse, male 33)*

> *"The providers are very friendly. . .that makes one feel like opening up." (Dental Assistant, Female, 42)*

The intervention team noted confidentiality as a major motivation behind health workers' motivation to seek services.

**Table 3. Prevalence of self-reported previous conditions and results from health checks among health workers in the ICAROZ comprehensive health check program; July 2020—June 2022.**

| Condition | Self-reported prevalence N = 6598 | Screened N | N positive on screening(%) |
|---|---|---|---|
| HIV | 675 (10%) | 3037 | 17 (0.6%) |
| SARS-CoV-2 infection | 1109 (17%) | 6598 | 149 (2.3%)[*1] |
| Tuberculosis | 2 (<0.1%) | 6598 | 0[*2] |
| Mental health condition / symptoms of common mental health disorder | 7 (0.1%) | 6598 | 645 (9.8%) |
| Anaemia | 6 (<0.1%) | 2723 | 335 (12%) |
| Diabetes / elevated HbA1c | 239 (3.6%) | 5532 | 428 (7.7%) |
| Hypertension / elevated blood pressure | 1,124 (17%) | 6598 | 1,249 (19%) |
| Renal disease / low eGFR | 1 (<0.1%) | 144 | 4 (2.8%) |
| Vision problem | 4 (<0.1%) | 4629 | 452 (9.8%) |
| BMI, kg/m$^2$ | | 6598 | |
| <18.5, underweight | | | 171 (2.6%) |
| 18.5–24.9, average | | | 2,259 (34%) |
| 25.0–29.9, overweight | | | 2,036 (31%) |
| ≥30.0, obese | | | 2,132 (32%) |
| Eligible for cervical cancer screening | | 5215 | 878 (17%) |
| Malignancy | 3 (<0.1%) | | |
| Epilepsy | 11 (0.2%) | | |
| Respiratory condition | 163 (2.5%) | | |
| Cardiovascular condition | 42 (0.6%) | | |

Abbreviations

BMI = Body Mass Index

eGFR = estimated Glomerular Filtration Rate

Foot notes

[*1] 918 health workers had symptoms suggestive of SARS-CoV-2, 877 were tested for SARS-CoV-2

[*2] 40 health workers had positive WHO TB symptom screening, 19 were tested for TB

*"All health workers in hospitals and clinics where we conducted the study were happy to be screened for various conditions by unfamiliar people. This made them feel comfortable and open up. Despite conducting Wellness clinics at their workplaces, they are not comfortable because they have no assurance of privacy and confidentiality." (Intervention staff, Nurse, Female, 47)*

Health workers cited several barriers to accessing the services. Lack of interest and time constraints hindered some accessing the services.

*"Mmm, time, waiting time is, is, is a lot. As staff we are inviting each other at once, inviting each other to go, about 10 of us at once so you will see as though. . . So, I waited about 10 minutes and I felt I was being delayed then I decided to go back. (Dental Assistant, Female, 42)*

*"People are afraid of being diagnosed because they think it's the end of life, they are in denial." (Youth Mentor, Male, 52)*

The intervention staff, highlighted points raised by health workers on the intervention's shortcoming. Not having doctors to confirm diagnosis and the lack of treatment services on

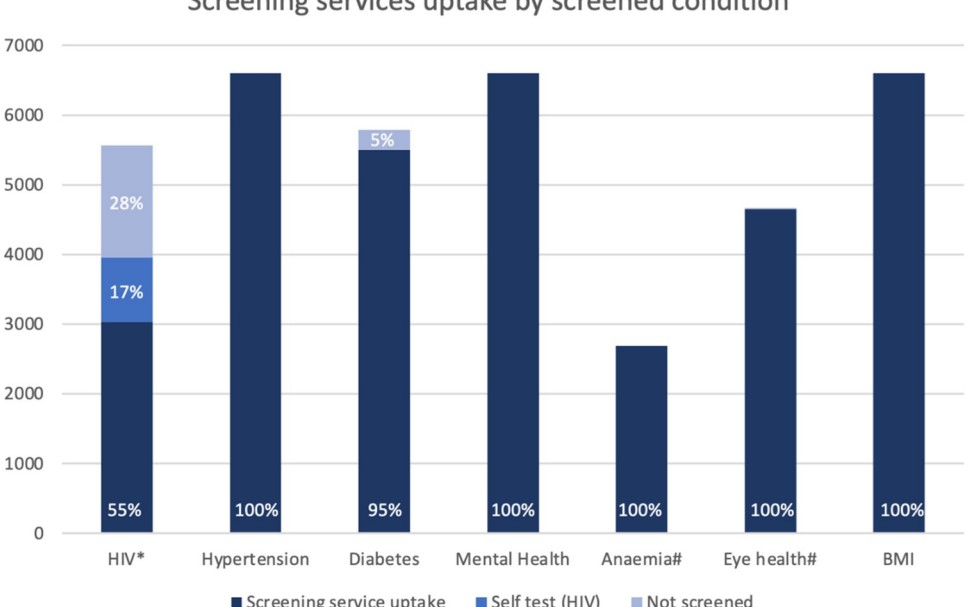

**Fig 4. Number of people eligible and taking up each of the screening tests offered.**

site was pointed out as a shortcoming especially for those health workers who could not afford to access further care.

> *"What they did not like about the intervention is they would say you are just providing testing services without treatment and we do not have the money to get treatment after being diagnosed"* (Intervention staff, Nurse, Male, 39)

Despite the barriers the intervention was perceived to have benefits for the health workers. Health workers were aware that their health seeking was poor due to competing priorities. They acknowledged that they often prioritised their work over managing their own health. Some were aware of pre-existing conditions and yet these were poorly managed. This was also highlighted by the intervention staff, who noted that health workers tend to neglect their own health needs. The health checks were believed to have promoted health awareness and the need to access health services.

> *"So, I have seen as health workers we tend to turn a blind eye when it comes to our health and say 'we will see'. Sometimes it's fear, but sometimes it is the lack of interest and ignorance."* (Nurse, Female, 30)

> *"I feel that it's crucial to always get screened because sometimes we might just be helping others while our health is deteriorating."* (Nurse, Female, 34)

> *"When implementing the intervention, I liked screening health workers for different diseases such as BP, diabetes etc. Health care workers are always busy nursing patients and they tend to forget themselves. The intervention made it possible for them to know their health status."* (Intervention staff, Nurse, Female, 47)

*"It was poor management of BP that pushed me. . . . . .I was diagnosed in 1998 and I was not taking my medication very well. I was not taking them as per prescribed combination."* (Youth Mentor, Male, 52)

*"I was also delighted that when we conducted mental health care screening, some of the health care workers who had issues were so glad to open up. They were happy to receive referral letters for further counselling services."* (Intervention staff, Nurse, Male, 39)

*"The services are very good because when I met them, I was a bit depressed and I was helped, I managed to get counselling, because the time I was helped, I was suicidal. Mental health was relevant to me because I was having problems. I had already given up. . .now I'm feeling better."* (Security, Female, 38)

Through provision of health check and hence diagnosis of previously unrecognised conditions, health workers thought that overall better control of chronic conditions would be achieved amongst health workers.

*"I also got a test for blood sugar but they said it was a bit high. They recommended me to go and see the doctor. . . .. I got to find out about it on the day I came for screening, I had never been diagnosed with diabetes before."* (Nurse, Female, 30)

*"The services were so fantastic; they will also help to improve disease management and prevention as far as health workers are concerned."* (Counsellor, Male, 29)

*"But I was not too surprised that my BP was high because there is a time it was borderline, but I was really happy to know because BP is a silent killer. Now that I know I can control my salt intake and have regular BP checks."* (Midwife, Female, 59)

### Adaptation of the service and recommendations for the future

The services offered evolved over time in response to feedback from clients and providers. Health workers recommended an expansion of the services to include conditions that were not part of the initial 'package' (see Fig 1) and to offer services repeatedly. They felt that while free-of-charge screening was a good start, follow-up care and treatment should also be provided to ensure maximum benefit.

*"Such a good service needs to expand (e.g., include STI, treatment services, training sessions on infection prevention) as well as to be well-staffed to meet our demand"* (Nurse, Female, 29)

*"Yeah, I was talking to them saying why don't you do, family planning services, I would have had one today"* (Nurse, Female, 34)

*"I think you need to do it maybe this after every 6 months or yearly."* (Facility manager, Female, 39)

*"I would recommend you make two visits a year at each facility."* (Nurse, Male, 33)

Several other recommendations to refine the services were highlighted by both the health workers and the intervention staff. Both intervention recipients and the providers felt that the service would be better if doctors, counsellors, and other specialists were part of the service provider team. In addition, they suggested that the services should extend to provision of treatment, for example medication or spectacles. The intervention staff also recommended

extending the reach to remote areas, where they believed some of the screening services would not be easily accessible for the health workers there.

*"The health care workers suggested that we should have a doctor on site so that after screening they can have consultation with him/her. They also suggested that we supply anti-hypertensive and anti-diabetes drugs." (Intervention staff, Research assistant, 36)*

*"Maybe on your team if you can have a doctor so that when you do your referrals, if someone is comfortable, they can then proceed into the doctor's office. Then they immediately see the doctor." (Nurse, Female, 30)*

*"If you test someone and see that they need medication. . .it's better that you give him the medication yourself. . . so that there is no gap. . . Also, to give eye glasses to those that have visual problems. They would want the glasses because the eye specialists are very expensive, they cost around $200." (Midwife, Female, 44)*

*"Future studies should have a team that includes a medical doctor, professional counsellors, anti-hypertensive drugs and oral diabetes drugs to do screening, diagnosis and initiate treatment to those eligible so that we can have fit healthcare workers." (Intervention staff, Nurse, Female, 47).*

*"Given a chance to provide this intervention again, I would recommend that we invest in extending the reach and offer the services to health workers who work in remote (rural) areas as they could have also benefited from the services such as the HBA1C testing which is not easily accessible and is expensive." (Intervention staff, Research assistant, Female, 30).*

## Discussion

Offering a comprehensive health check to health workers during working hours was feasible and acceptable, not just for the clients but also for their health facility managers. Among those who took up the service, uptake of each service component was almost 100%, except for HIV, suggesting very high acceptability. The burden of previously unrecognised chronic conditions was high, illustrating the unmet need for chronic disease services among health workers in Zimbabwe.

The service evolved over time with additional screening modalities being added, the geographical reach extended, and delivery methods refined. Given the rapidly changing situation during the SARS-CoV-2 pandemic, flexible service provision and adjustments were key. There were certain 'ingredients' which were important to make the service accessible and acceptable: health checks being i) free, ii) comprehensive, iii) efficient, and iv) offered by professional and friendly staff. As this intervention was designed as a one-off health check, it was not possible to offer on-site treatment services because continuity of care could not be maintained. Given the limited treatment services available for non-communicable diseases in Zimbabwe, health workers repeatedly highlighted the need for continuous services integrated or co-located with treatment and care. This would support further development of this approach as a holistic occupational health service for health workers, integrated into health facilities and providing ongoing, on-site diagnostic and treatment services. In such an intervention, ensuring confidentiality is maintained will be crucial for health workers to be willing to access the service.

In this study, very few new HIV diagnoses (n = 17) were made among health workers. Notably, HIV testing is the only screening, diagnostic, and treatment service available free of charge to anybody at any public health facility in Zimbabwe [33]. The most recent population-based HIV impact survey conducted in 2020 showed that 87% of adults living with HIV in

Zimbabwe know their HIV status and 97% of those who have received an HIV diagnosis are on antiretroviral therapy [34]. This may be attributed to a highly effective, vertical HIV programme [35] and is in stark contrast to the high proportion of undiagnosed hypertension and diabetes observed among health workers, perhaps reflecting the fact that treatment for these conditions is neither readily available in public health facilities nor affordable [36]. We also found that almost one in ten health workers had previously undetected vision impairment: addressing visual problems are important for safe patient care.

Health workers accessing the services were predominantly middle-aged to older female nurses. Women are generally more likely to seek health services than men [37–40] and thus female staff may have preferentially accessed the service. Older health workers may have felt that they were benefitting more from accessing the service given that some of the screened conditions such as hypertension, diabetes, and near vision impairment are more prevalent among older age groups. If this is the case, future development of occupational health services should seek to explore strategies that attract men and younger people to attend the service to ensure that they are not missed.

However, while we do not have age- and sex-stratified numbers of health workers employed at each of the facilities and hence cannot calculate uptake, recent reports show that women make up the majority of health professionals globally [41]. In Zimbabwe 76% of nurses are women [42], almost equal to the proportion of women who accessed the health check-up services. The global health workforce has aged significantly over the past decade [43, 44], and increasing age is associated with increased risk of many chronic conditions including diabetes, hypertension and vision impairment. In fact, 0.3% of health workers accessing the service in this study were older than 65 years, which would be the normal retirement age in Zimbabwe and the median age of health workers accessing the service was 37 which is older compared to the general population (median age 18) [45]. It is likely that a combination of young health workers leaving the country and inflation devaluing pensions are forcing health workers to work longer. This is another important justification for the development and implementation of occupational health services: keeping health workers healthy by diagnosing and treating chronic conditions, as well as preventing both chronic conditions and their complications is crucial to enable them to continue providing health services as they age.

Community SARS-CoV-2 infection rates had an impact on service uptake and delivery, which should be considered for future development of occupational health services. There was particularly high demand for the service during SARS-CoV-2 waves; however, SARS-CoV-2 infections among the team and limited operating hours resulted in challenges in meeting this demand. Strategies to ensure service provision was maintained included implementing task shifting, reduced lunch break time with packed lunch delivery, and re-deployment of staff from other projects so as to increase efficiency of the service. These approaches may also be useful for maintaining service delivery in other types of crises.

This evaluation of the health check service has several strengths. The sample size was large both with regards to health workers and health facilities, with the latter including a diverse range of health facilities from tertiary to primary level and across different provinces. In addition, it was conducted in "real-world settings" during a pandemic demonstrating generalisability and scalability in challenging contexts. Similar prevalence of undiagnosed conditions have been reported from other (non-health worker) settings in Africa, and it is likely that our intervention is generalisable elsewhere [46–48]. A detailed process evaluation was conducted with real-time analysis to inform refinement of the service. The service was an innovation which manged to address several unmet needs of health workers. It was free, accessible, and comprehensive, facilitating uptake, and included education to improve health awareness. Follow up calls to health workers testing SARS-CoV-2 positive at day 1, 3, 7 and 10 after accessing the

service and extending free SARS-CoV-2 testing to family members with respiratory symptoms acted as psychosocial support to health workers in a period of need [28].

We acknowledge several limitations. The findings of this study cannot be generalised to all health workers working at the included facilities because of likely self-selection bias. Equally selection with regards to health facilities is highly likely because facilities were selected on the basis of accessibility and logistics. Hence more decentralised and rural health facilities were not included in the study, which means that results cannot be generalised to these settings. In addition, social desirability bias may have influenced the results of the process evaluation. Health workers may not have felt free to discuss challenges encountered when accessing the health service.

The overall uptake of the service could not be determined as denominators (number of health workers eligible) were not available. Regarding screening procedures, blood pressure was measured on a single visit (albeit with measures done according to WHO guidelines). Repeat blood pressure measurement may have reduced the number of people classified as having high blood pressure [49]. HbA1c cut-point of 6.5% was used to screen for diabetes. However, this threshold has been shown to have low sensitivity in African populations [50, 51], hence the prevalence of diabetes may have been underestimated. Uptake of HIV testing among people who had not tested in the past three months was incomplete, potentially meaning that people with undiagnosed HIV were missed. This may reflect participating health workers opting to test elsewhere (given that HIV testing is widely available and free, unlike the other services offered) however, may also reflect a reluctance to test for HIV in a workplace-based setting due to concerns about confidentiality and stigma. In this study, we did not evaluate the cost of service provision, something that is needed to inform future healthcare planning, and this could recommended as an idea for future study.

In conclusion, we implemented a comprehensive health check which was highly acceptable, with results demonstrating a considerable unmet need for chronic disease prevention, screening, and treatment among health workers in Zimbabwe. Whilst there were many interventions, globally, that provided SARS-CoV-2 testing or mental health support for healthcare workers during the COVID-19 pandemic [52], we are not aware of any other comprehensive health services developed for health workers during this period. Obtained results from the service provide a strong justification to implement continuous occupational health programmes for health workers which go beyond conventional occupational risk (i.e., air- and blood-borne infections) and address health workers' health and wellbeing more holistically. Free, accessible, and comprehensive services are vital to foster a shift from a reactive health response to a proactive health seeking and prevention culture.

## Supporting information

**S1 Fig. Design used for the poster and fliers.**
(TIFF)

**S1 Table. Process evaluation qualitative reports (Themes, codes and supporting quotations from workshops).**
(DOCX)

**S1 File. Inclusivity in global research questionnaire.**
(DOCX)

## Acknowledgments

We acknowledge the support received from the Global Public Health strand of the Elizabeth Blackwell Institute for Health Research, Sheffield and Oxford universities, Wellcome Trust,

UK Medical Research Council (MRC) and the UK Foreign, Commonwealth and Development Office (FCDO), the UK government and the government of Canada.

## Author Contributions

**Conceptualization:** Edson T. Marambire, Rudo M. S. Chingono, Grace McHugh, Celia L. Gregson, Katharina Kranzer.

**Data curation:** Claire J. Calderwood, Leyla Larsson, Tsitsi Bandason, Nicol Redzo.

**Formal analysis:** Claire J. Calderwood, Leyla Larsson, Victoria Simms.

**Funding acquisition:** Victoria Simms, Celia L. Gregson, Rashida A. Ferrand, Katharina Kranzer.

**Investigation:** Victoria Simms, Celia L. Gregson, Chiratidzo E. Ndhlovu, Hilda Mujuru, Simbarashe Rusakaniko, Rashida A. Ferrand, Katharina Kranzer.

**Methodology:** Sibusisiwe Sibanda, Chiratidzo E. Ndhlovu, Hilda Mujuru, Simbarashe Rusakaniko, Rashida A. Ferrand, Katharina Kranzer.

**Project administration:** Edson T. Marambire, Rudo M. S. Chingono, Fungai Kavenga, Farirai P. Nzvere, Grace McHugh.

**Resources:** Aspect J. V. Maunganidze, Christopher Pasi, Michael Chiwanga, Prosper Chonzi.

**Supervision:** Rashida A. Ferrand, Katharina Kranzer.

**Validation:** Rashida A. Ferrand, Katharina Kranzer.

**Visualization:** Ioana D. Olaru.

**Writing – original draft:** Edson T. Marambire, Rudo M. S. Chingono.

**Writing – review & editing:** Claire J. Calderwood, Leyla Larsson, Farirai P. Nzvere, Ioana D. Olaru, Victoria Simms, Celia L. Gregson, Rashida A. Ferrand, Katharina Kranzer.

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
