## [Decision Letter · Decision Letter 0]

7 Sep 2023

PGPH-D-23-01432

Evaluation of a comprehensive health check offered to frontline health workers in Zimbabwe

Dear Dr. Marambire,

Thank you for submitting your manuscript to PLOS Global Public Health. After careful consideration, we feel that it has merit but does not fully meet PLOS Global Public Health’s publication criteria as it currently stands. Therefore, we invite you to submit a revised version of the manuscript that addresses the points raised during the review process.

We look forward to receiving your revised manuscript.

Kind regards,

Carmen S. Christian, PhD

Academic Editor

Journal Requirements:

Additional Editor Comments (if provided):

Reviewers' comments:

Reviewer's Responses to Questions

**Comments to the Author**

1. Does this manuscript meet PLOS Global Public Health’s publication criteria? Is the manuscript technically sound, and do the data support the conclusions? The manuscript must describe methodologically and ethically rigorous research with conclusions that are appropriately drawn based on the data presented.

Reviewer #1: Yes

Reviewer #2: Yes

2. Has the statistical analysis been performed appropriately and rigorously?

Reviewer #1: Yes

Reviewer #2: Yes

3. Have the authors made all data underlying the findings in their manuscript fully available (please refer to the Data Availability Statement at the start of the manuscript PDF file)?

Reviewer #1: Yes

Reviewer #2: Yes

4. Is the manuscript presented in an intelligible fashion and written in standard English?

Reviewer #1: Yes

Reviewer #2: Yes

5. Review Comments to the Author

Reviewer #1: Manuscript Number PGPH-D-23-01432

Reviewer Comments

1. This is an original well-designed, relevant, timely and well-reported study on the implementation and uptake of a health check service for Zimbabwe’s health care workers during the COVID-19 pandemic, 2020 – 2022. The health care deficit addressed is explained clearly.

The best attributes of this paper are the clearly explained and justified methodology and approaches used during the evaluation. In the results, the detection of undiagnosed hypertension and diabetes was/is a very valuable aspect of this study and any future follow up programmes. Even if treatment is expensive or not available, awareness is important and life style changes can make a different in some cases.

2. However, there is room for improvement in the following:

2.1 Legal, Ethical and data security issues. Lines 249 – 255 summarise the ethical approvals.

2.1.1 Other than the informed consent, the other ethical, legal and data protection issues that arise/arose in this study and the mitigation measures taken are unclear.

2.1.2 What happens (ed) to the recordings and transcripts? How did the study manage these so as to guarantee security, confidentiality, privacy and security during and after the study?

2.1.3 The authors are multi-sited and this implies that data may have been transmitted across national territories. If this was/is the case, what mechanisms were employed to share data across boundaries and institutions and what measures were used to ensure data security?

There are some small points for clarification, viz:

2.2 Lines 242 – 246 – in the methods section on coding, the study identified deductive and inductive themes. However, the specific themes identified for each category and how they were used to organise the presentation of results is unclear.

2.3 The target was health workers. It is unclear why and how police and army personnel became part of the study as reported in the results (lines 271 -274).

2.4 The study reports on age, sex distribution of clients (lines 270 – 271) indicating that 79% of them were female. How do the proportions reported in the study compare to those in the total national or provincial health workforce? This issue comes up again later in lines 444 -451 (page 25) where it states that 76% of Zimbabwe’s nurses are female. The 3% difference suggests that the proportions in the study come close to those in the national workforce. It is surprising that the stratified numbers/percentages in the total workforce are not available from the relevant ministry.

2.5 Reference is needed for the claim made on “… low sensitivity in African populations …” at page 27 lines 494 – 496.

3. Further attention is needed to address a few errors of grammar.

4. One would expect spatial variation in the health conditions, experiences and service expectations among the clients; remote rural centres compared to the metropolitan areas for instance. If the study did not have scope to reflect on this, then this could also be recommended as a candidate for inclusion in future studies.

Recommended for publication with minor corrections. END

Reviewer #2: This is a good paper about a relevant and important topic especially given how the well-being of health workers is often overlooked. Please see below my suggestions to help strengthen the manuscript.

INTRODUCTION

The authors may want to include literature that shows the conditions/ problems that typically affect health workers to justify the need for health checks.

METHODS

I suggest placing the “Health check” section [line 166-189] before the “Service setup section” [line 144-162]. This will help the manuscript read better by first describing the intervention and then how it Is set up.

RESULTS

• Not sure if some of the referrals were followed up to see the uptake of referral service? It would be beneficial to know the proportion of those who utilized the referral services.

• In the methodology, the authors mention that they interviewed service providers (assuming these were the delivering agents on the intervention). However, in the results section, their views are not present or have not been distinctly labelled. It would strengthen the manuscripts to also include their experiences delivering the intervention and improvements that might need to be made to the intervention.

• It would be beneficial to hear their findings on sustainability from the service providers (e.g., time, workload).

DISCUSSION

The authors did not discuss how their intervention compares to other health workers/ frontline worker interventions that were implemented whether before or during the COVID outbreak.

6. PLOS authors have the option to publish the peer review history of their article (what does this mean?). If published, this will include your full peer review and any attached files.

**Do you want your identity to be public for this peer review?** For information about this choice, including consent withdrawal, please see our Privacy Policy.

Reviewer #1: **Yes: **Beacon Mbiba

Reviewer #2: No

---

## [Editor Report · Decision Letter 1]

28 Nov 2023

PGPH-D-23-01432R1

Evaluation of a comprehensive health check offered to frontline health workers in Zimbabwe

Dear Dr. Marambire,

Thank you for submitting your revised manuscript to PLOS Global Public Health. After careful consideration, we feel that it has merit but does not fully meet PLOS Global Public Health’s publication criteria as it currently stands. Therefore, we invite you to submit a second revised version of the manuscript that addresses the point raised below.

Currently, the manuscript does not include a discussion on potential sources of bias. Please address this in the second revised version.

We recommend that authors use the COREQ checklist, or other relevant checklists listed by the Equator Network, such as the SRQR, to ensure complete reporting (http://journals.plos.org/globalpublichealth/s/submission-guidelines#loc-qualitative-research). In general, we would expect qualitative studies to include the following: 1) defined objectives or research questions; 2) description of the sampling strategy, including rationale for the recruitment method, participant inclusion/exclusion criteria and the number of participants recruited; 3) detailed reporting of the data collection procedures; 4) data analysis procedures described in sufficient detail to enable replication; 5) a discussion of potential sources of bias; and 6) a discussion of limitations.

A rebuttal letter that responds to the point raised above by the editor. You should upload this letter as a separate file labeled 'Response to Editor'.A marked-up copy of your manuscript that highlights changes made to the original version. You should upload this as a separate file labeled 'Revised Manuscript with Track Changes'.An unmarked version of your revised paper without tracked changes. You should upload this as a separate file labeled 'Manuscript'.

We look forward to receiving your revised manuscript.

Kind regards,

Carmen S. Christian, PhD

Academic Editor

Journal Requirements:

2. We do not publish any copyright or trademark symbols that usually accompany proprietary names, eg  ©, ®, ™  (e.g. next to drug or reagent names). Please remove all instances of trademark/copyright symbols throughout the text, including ® on page 12.

3. Some material included in your submission may be copyrighted. According to PLOS’s copyright policy, authors who use figures or other material (e.g., graphics, clipart, maps) from another author or copyright holder must demonstrate or obtain permission to publish this material under the Creative Commons Attribution 4.0 International (CC BY 4.0) License used by PLOS journals. Please closely review the details of PLOS’s copyright requirements here: PLOS Licenses and Copyright. If you need to request permissions from a copyright holder, you may use PLOS's Copyright Content Permission form.

Potential Copyright Issues:

Fig 3: please (a) provide a direct link to the base layer of the map (i.e., the country or region border shape) and ensure this is also included in the figure legend; and (b) provide a link to the terms of use / license information for the base layer image or shapefile. We cannot publish proprietary or copyrighted maps (e.g. Google Maps, Mapquest) and the terms of use for your map base layer must be compatible with our CC-BY 4.0 license. 

"

Additional Editor Comments (if provided):

None

Reviewers' comments:

None

---

## [Editor Report · Decision Letter 2]

5 Dec 2023

Evaluation of a comprehensive health check offered to frontline health workers in Zimbabwe

PGPH-D-23-01432R2

Dear Mr Marambire,

We are pleased to inform you that your manuscript 'Evaluation of a comprehensive health check offered to frontline health workers in Zimbabwe' has been provisionally accepted for publication in PLOS Global Public Health.

Best regards,

Carmen S. Christian, PhD

Academic Editor